# A Traffic Light System to Maximize Carbohydrate Cryoprotectants’ Effectivity in Nanostructured Lipid Carriers’ Lyophilization

**DOI:** 10.3390/pharmaceutics13091330

**Published:** 2021-08-25

**Authors:** Helena Rouco, Patricia Diaz-Rodriguez, Alba Guillin, Carmen Remuñán-López, Mariana Landin

**Affiliations:** 1R+D Pharma Group (GI-1645), Strategic Grouping in Materials (AEMAT), Department of Pharmacology, Pharmacy and Pharmaceutical Technology, Campus Vida, Faculty of Pharmacy, Universidade de Santiago de Compostela, 15782 Santiago de Compostela, Spain; helena.rouco@rai.usc.es (H.R.); alba.gl96@gmail.com (A.G.); 2Drug Delivery Systems Group, Department of Chemical Engineering and Pharmaceutical Technology, Campus de Anchieta, School of Sciences, Universidad de La Laguna (ULL), 38200 La Laguna, Spain; 3NanoBiofar Group (GI-1643), Department of Pharmacology, Pharmacy and Pharmaceutical Technology, Campus Vida, Faculty of Pharmacy, Universidade de Santiago de Compostela, 15782 Santiago de Compostela, Spain; mdelcarmen.remunan@usc.es

**Keywords:** artificial intelligence, cryoprotectants, lyophilization, nanostructured lipid carriers, neurofuzzy logic, sugars, traffic light system

## Abstract

Lyophilization is often employed to transform nanoparticle suspensions to stable solid forms. This work proposed Neurofuzzy Logic (NFL) to better understand the lyophilization process of Nanostructured Lipid Carriers’ (NLCs) dispersions and the carbohydrate cryoprotectants’ (CPs) performance in these processes. NLCs were produced by hot homogenization, frozen at different speeds, and lyophilized using several CPs at variable concentrations. NLCs were characterized, and results were expressed as increase in particle size (Δ size), polydispersity (Δ PdI), and zeta potential (Δ ZP) of lyophilized powders (LP) regarding initial dispersions. CPs were classified according to their molecular weights (MW), and the osmolarities (Π) of CPs solutions were also determined. Databases obtained were finally modelled through FormRules^®^ (Intelligensys Ltd., Kirkwall, Scotland, UK), an NFL software. NFL models revealed that CPs’ MW determines the optimal freezing conditions and CPs’ proportions. The knowledge generated allowed the establishment of a traffic light system intended to successfully select and apply sugars for nanoparticles lyophilization.

## 1. Introduction

Nanostructured Lipid Carriers (NLCs) are drug delivery systems experiencing increased attention in the pharmaceutical field [1]. They exhibit a core matrix composed of solid and liquid lipids, showing several advantages over conventional carriers, such as improved solubility capacity and drug half-life, greater permeability, and better stability during storage [1]. Furthermore, in recent years, scientific literature has evidenced the utility of these nanocarriers to achieve controlled release after intravenous [2], pulmonary [3], oral [4], or topical [5] administration.

Despite these promising features, lipid nano-dispersions are susceptible to hydrolysis, gelation, flocculation, creaming, and sedimentation or coalescence, triggering system destabilization [6,7,8]. These stability issues, together with the need to produce easy-to-transport-and-store dosage forms [9], make necessary their drying to obtain a powder that, once reconstituted, generates the original NLC. With this regard, lyophilization using cryoprotectants (CPs) was considered an appropriate approach to enhance long-term nanoparticles’ stability, avoiding freezing stress [8,10]. This technique has gained relevance nowadays, due to the urgent need of improving the long-term stability of certain lipid nanoparticle-based formulations, as is the case of SARS-CoV-2 vaccines [11].

Carbohydrate cryoprotectants (CPs), such as trehalose, mannitol, sucrose, or glucose, are the most popular cryoprotectants for nanoparticle lyophilization [10,12,13,14]. These compounds are known to vitrify at a certain temperature, generating a glassy matrix capable of protecting nanoparticles from the ice mechanical stress [7,15].

The lyophilization performance involves a significant number of variables related to the characteristics of the formulation, the addition or not of cryoprotectants, their type and content, and the operating conditions [7]. In general, lyophilization conditions are established by trial-and-error procedures, without considering the interactions between these variables, or analyzing in depth the mechanisms involved in the process [10].

In the last decade, Artificial Intelligence (AI) tools, such as Artificial Neural Networks (ANNs) or Neurofuzzy Logic (NFL) systems, have gained increasing attention for pharmaceutical applications [16] being used to study and optimize various processes such as wet granulation [17], the formulation of micro- and nanoparticles [18,19], or the preparation of hydrogels [20,21]. In these systems, AI allowed the prediction of the endpoint of the granulation process, to obtain particulate drug delivery systems exhibiting optimal characteristics, and also to develop smart, thermosensitive hydrogels [17,18,19,20,21].

ANNs are biologically inspired artificial intelligence tools, designed to mimic the information processing of the human brain, allowing the establishment of relationships between the process variables (inputs) and experimental results (outputs) [20]. The main unit of an artificial neural system is the artificial neuron or node. These nodes establish connections with each other and the strength of these connections is known as weight [22]. However the interpretation of these ANNs might not be a simple task [23]. NFL systems are hybrid technologies remarkably suitable for data mining, as they integrate the ability of ANNs to learn from data and the capacity of fuzzy logic to express concepts in a simple way through linguistic “IF-THEN” rules [16].

Therefore, the main hypothesis of this work was that AI tools, such as NFL, could be useful to rationally establish a suitable lyophilization procedure for NLCs using carbohydrates as cryoprotectants, a procedure traditionally established experimentally. Furthermore, this information would predictably allow achieving a better understanding of the lyophilization conditions’ (CP type and concentration or freezing speed) impact on lyophilized NLCs’ properties, and also to assess the physicochemical phenomena driving sugars’ effectivity as CP.

## 2. Materials and Methods

### 2.1. Materials

Precirol^®^ ATO 5 (glyceryl distearate) and Epikuron^®^ 145V (deoiled phosphatidyl choline-enriched lecithin) were kindly donated by Gattefossé (Saint-Priest, France) and Cargill (Wayzata, MN, USA), respectively. Polysorbate 80 (Tween^®^ 80), Oleic acid, D-(+)-trehalose dihydrate, D-mannitol (≥98%), D-(−)-fructose, D-sorbitol, and dialysis membrane (Spectrum™ Labs Spectra/Por, MWCO 3.5 KDa) were acquired from Sigma Aldrich (St Louis, MO, USA). D-glucose anhydrous and lactose were obtained from Fisher Scientific (Hampton, NH, USA) and Merck (Madrid, Spain), respectively. D-(+)-sucrose was acquired from Acros Organics, Fisher Scientific (Hampton, NH, USA). Ultrapure water (MilliQ plus, Millipore Ibérica, Madrid, Spain) was used throughout.

### 2.2. Methods

#### 2.2.1. NLCs’ Formulation

NLCs’ formulations were prepared by hot, high-shear homogenization, following the procedure and composition previously optimized in our laboratory through Artificial Intelligence tools [19]. Oleic acid and Precirol^®^ ATO 5 (Gattefossé, Saint-Priest, France) were used as liquid and solid lipid, respectively, while Epikuron^®^ 145V (Cargill, Wayzata, MA, USA) and Tween^®^ 80 (Sigma Aldrich, St. Louis, MO, USA) were employed as emulsifiers. Briefly, an oil phase (300 mg), composed by a 75:25 ratio of liquid:solid lipid, was melted at 80 °C. Then, an aqueous phase (10 mL), comprising Epikuron^®^ 145V (Cargill, Wayzata, MA, USA) (0.5% w/w regarding oil phase) and Tween^®^ 80 (Sigma Aldrich, St. Louis, MO, USA) (1.9% *w/v* regarding aqueous phase), was also heated at 80 °C and added to the oil phase. The mixture was hot, high-shear homogenized at 14,800 rpm for 10 min using an Ultra-Turrax T25 (IKA Labortechnik, Staufen, Germany), leading to a NLC dispersion, which was cooled in an ice bath for 2 min with gentle stirring.

Furthermore, to remove the non-incorporated components, nanoparticle dispersions were dialyzed overnight using a porous membrane (Spectrum™ Labs Spectra/Por, MWCO 3.5 KDa) (Sigma Aldrich, St. Louis, MO, USA). The particle size and surface charge of the NLCs were characterized as indicated below.

#### 2.2.2. NLCs’ Lyophilization and Reconstitution

A selection of carbohydrate cryoprotectants (trehalose, lactose, sucrose, sorbitol, glucose, fructose, and mannitol) at different concentrations (2.5, 5, 10, 15, and 20% *w/v*) were used for the lyophilization of NLCs. In addition, the effect of two freezing procedures was also evaluated (fast by immersion in liquid nitrogen or slow in a freezer at −80 °C).

Briefly, 2 mL of dialyzed NLC suspensions were mixed with accurate amounts of the CPs in 5-mL tubes. Then, mixtures were manually homogenized (by tube inversion) until complete dissolution and subsequently frozen. Lyophilization was carried out by duplicate in a freeze dryer Telstar LyoQuest Plus −85 °C/ECO (Telstar, Madrid, Spain) for 24 h. During the process, the chamber temperature was maintained at −70 °C, under a high vacuum of 0.01 mbar, approximately.

Lyophilized NLCs (50 mg) were resuspended in 10 mL of Milli-Q^®^ (Millipore Ibérica, Madrid, Spain ) water. The dispersions were shaken manually and then sonicated for 30 s at a frequency of 20 kHz with a Sonicator 700W Sonic Dismembrator (Fisher Scientific, Hampton, NH, USA) to ensure complete resuspension. Then, the particle size and surface charge of the NLCs were again characterized.

#### 2.2.3. Particle Size and Surface Charge Characterization

NLCs’ particle size, polydispersity index (PdI), and surface charge, before and after lyophilization, were determined using a Zetasizer Nano ZS (Malvern Instruments, Malvern, UK). Particle size and PdI measurements were performed in polystyrene cuvettes, after proper dilution with Milli-Q^®^ (Millipore Ibérica, Madrid, Spain ) water. Surface charge was determined through particle mobility in an electric field as zeta potential (ZP). For this purpose, a specific cuvette was employed where a potential of ± 150 mV was established. All measurements were conducted by triplicate at 25 ± 1 °C. Results were expressed as the increase in particle size (Δ size), polydispersity index (Δ PdI), and zeta potential (Δ ZP).

#### 2.2.4. Osmolarity Determination

Samples of Milli-Q^®^ (Millipore Ibérica, Madrid, Spain ) water, NLCs’ dispersions, and aqueous solutions of the CPs under evaluation were experimentally evaluated using a Vapro^®^ vapor pressure osmometer (model 5600, Wescor, ELITechGroup, Logan, UT, USA). Osmolarity values of CPs’ solutions were determined using concentrations ranging between 2.5–20%, except for lactose and mannitol, where the studied concentrations were 2.5–10% or 2.5–15%, respectively, due to their low aqueous solubility.

#### 2.2.5. Modelling through Artificial Intelligence Tools

The variables studied followed an experimental design for three variables (CP selected, CP concentration, and freezing speed) at 7, 5, and 2 levels, respectively. Additionally, the database was completed with the CPs’ molecular weight (MW_CP_) and the osmolarity at the specific concentration (Table 1). Moreover, the parameters derived from the NLCs’ analysis regarding particle size and surface charge expressed as Δ Size, Δ PdI, and Δ ZP were also added to the database.

The complete database was modelled using FormRules^®^ v4.03 (Intelligensys Ltd., Kirkwall, Scotland, UK), which is a Neurofuzzy Logic (NFL) software that enables answering “WHAT IF” questions through the generation of “IF-THEN” rules [19]. “IF-THEN” rules consist of an antecedent and a consequent part, indicating the relationship between the variables or inputs and the resulting values or outputs [19]. These rules were obtained after a fuzzification process, where each value of an input was classified and described by a word (low, medium, or high) and an associated membership degree (MD) ranging from 0 to 1. Values of MD close to 1 indicated a certain hypothesis was true (e.g., Δ Size is low), while values of MD close to 0 imply a certain hypothesis was false (e.g., Δ Size is not low) [16].

Two Neurofuzzy logic models were carried out. Model 1 studied the effect of freezing speed, the CP type, and CP percentage (included as inputs), on the Δ Size, Δ PdI, and Δ ZP of the lyophilized powders (LPs, included as outputs). Neurofuzzy logic systems need to be trained to learn from data and, during this training process, they established relationships between the inputs and the outputs of the database using different algorithms to alter the strength of the connections in the neural network [22]. This process allowed us to modulate the signal flow, establishing generalizations between the variables studied (inputs) and the nanoparticle characteristics (outputs) [22]. In this way, the training parameters selected for model 1 were: ridge regression factor of 1 × e^−6^, two set densities, Structural Risk Minimization as model selection criteria (C_1_ = 0.70 and C_2_ = 4.80), two maximum inputs per submodel, and 15 maximum nodes per input.

Model 2 explored the effect of specific CP characteristics (CP type, MW_CP_, osmolarity) and the freezing speed on the three outputs. For model 2, training parameters similar to model 1 were used, with two exceptions: C_1_ value needed adjustment (C_1_ = 0.928) and the maximum number of inputs per submodel increased to 4.

The quality of the models for each output was assessed using the calculated f ratio from the analysis of variance (ANOVA) and the determination coefficient of train set (R^2^), which estimated their accuracy and predictability, respectively. The train set R^2^ values were calculated as follows [24]:(1)$$R^{2}=[\;1-\overset{n}{\underset{i=1}{\sum }}{\left(yi-yi^{\prime }\right)}^{2}/\overset{n}{\underset{i=1}{\sum }}{\left(yi-yi\prime \prime \right)}^{2}]\times 100\%,$$
where *yi* is the experimental value obtained for a given output, *yi*′ is the predicted value for the output calculated by the model, and *yi*″ is the mean of the experimental value. Values of R^2^ ranging from 70% to 99.9% indicate satisfactory model predictabilities [25].

A calculated f ratio higher than the critical f value for the same degrees of freedom indicates there are not statistically significant differences between predicted and experimental data; therefore, the model is accurate.

## 3. Results

### 3.1. Cryoprotectants’ Characterization

The carbohydrate cryoprotectants (CPs) selected for this work included monosaccharides, such as fructose and glucose, disaccharides, such as lactose, sucrose, and trehalose, and sugar alcohols, such as mannitol and sorbitol (Figure 1). To better understand the CPs’ properties’ effect on the lyophilization procedure, these compounds were classified by their MW_CP_ and characterized in terms of osmolarity (Π) (Table 1). Fructose and glucose had a MW_CP_ of ≈180.16 g/mol, while lactose, sucrose, and trehalose had a MW_CP_ of ≈342.3 g/mol, and mannitol and sorbitol had a MW_CP_ of ≈182.17 g/mol [26].

On the other hand, Π led to values ranging from 154–1141, 135–1127, 72–248, 142–945, 131–1146, 73–602, and 70–538 mmol/kg for fructose, glucose, lactose, mannitol, sorbitol, sucrose, and trehalose, respectively, for the different concentrations under study. Π values were in agreement with those expected by multiplying CPs’ molar concentration by their dissociation factor, which is known to be 1 for molecules that do not dissociate in solution, as is the case for sugars. As an example, a 5% (*w/v*) solution of trehalose and glucose would exhibit a molarity of approximately 140 and 278 mmol/L, respectively, which closely agrees with the experimental Π values obtained.

### 3.2. Physicochemical Characterization of NLCs and Lyophilized Powders

Both size and PdI are key parameters determining nanoparticles’ stability and, therefore, it is of utmost importance to control Δ Size and Δ PdI of NLCs, which should be maintained as low as possible. NLC formulations (*n* = 5) showed initially a particle size of 126 ± 19 nm, a PdI of 0.28 ± 0.05, and a ZP of −26 ± 3 mV. The values of Δ Size, Δ PdI, and Δ ZP were 62 ± 9 nm, 0.10 ± 0.06, and −2 ± 1 mV for reconstituted NLCs when fast freezing was carried out and 61 ± 7 nm, 0.14 ± 0.07, and −1 ± 2 mV when slow freezing was performed. The nanocarriers exhibited a suitable ability to endure lyophilization without CPs. However, NLCs showed a gummy-like appearance, and some difficulties were found during the re-constitution step, which justified the need of CPs. The use of CPs allowed us to obtain nanoparticles with a dry appearance and easy and quick reconstitution. Moreover, after CPs’ incorporation, Δ Size values were in the range of 23–94 nm and 31–157 nm for fast and slow freezing, respectively. As can be observed in Figure 2, CPs’ effectivity widely varied over the range of concentrations tested. Furthermore, a different behavior pattern can be noticed as a function of the type of CP (monosaccharides, sugar alcohols, or disaccharides), the concentration used, and the freezing process. As an example, sucrose (a disaccharide) appeared to perform better at high concentrations (Figure 2A,E). At 2.5%, and with fast freezing, sucrose led a Δ Size of 80 ± 16 nm, while 20% sucrose promoted a smaller Δ Size of 40 ± 21 nm. In contrast, a monosaccharide such as fructose appeared to behave better at low proportions, exhibiting Δ Size values of 58 ± 2 nm and 87 ± 14 nm for concentrations of 2.5% and 20%, respectively, when fast freezing was employed. Additionally, as a general trend, the use of fast freezing seemed to favor low Δ Size values.

The Δ PdI values ranged from 0.06–0.37 and 0.09–0.51 for fast and slow freezing, respectively. Similar trends to those described for Δ Size in terms of CP type, concentration, and freezing speed were observed (Figure 3).

In contrast, a narrower results’ range of −5–+5 mV and −4–+7 mV was reported for Δ ZP using fast or slow freezing, respectively (Figure 4). In this case, positive increases were considered to be more relevant than negative ones, due to the impact that they could have over colloidal stability. However, only slight neutralizations were found with some CPs, such as lactose, fructose, or trehalose.

### 3.3. Influence of Lyophilization Variables over NLCs’ Characteristics (Model 1)

FormRules^®^ (Intelligensys Ltd., Kirkwall, Scotland, UK) succeeded in modelling Δ Size and Δ PdI parameters, as both R^2^ values were above 70% (Table 2), indicating suitable predictabilities [25]. Moreover, computed f values were higher than the critical ones for the degrees of freedom of the model, indicative of no statistically significant differences among predicted and experimental results and, therefore, accuracy [18]. However, limited predictability (R^2^ = 51.35%) was achieved for the Δ ZP model, probably due to the similarity of the values obtained, indicating ZP variations cannot be completely explained by the variables studied.

Both Δ Size and Δ PdI were affected by the three variables studied (freezing speed, CP selected, and concentration (%CP)), with the interaction between the CP and the CP concentration having the strongest effect on both outputs (highlighted submodels in Table 2).

According to the “IF-THEN” rules generated by FormRules^®^ (Intelligensys Ltd., Kirkwall, Scotland, UK) for Δ Size and Δ PdI (Appendix A), every CP required a specific range of concentration for its best performance. As a general trend, a fast freezing speed favored the obtention of formulations exhibiting a low Δ Size. The sugar alcohols, mannitol and sorbitol, led the worst results for Δ Size in the whole range of concentrations, using either fast or slow freezing speeds (Rules 19–30, Appendix A). Fructose and glucose required low-medium concentrations (up to 12.5%), along with a fast freezing speed (Rules 1–12, Appendix A). On the other hand, sucrose should be employed at a medium-high concentration (over 2.5%) if a fast freezing speed is employed. Furthermore, an even higher sucrose proportion (above 12.5%) would be required if slow freezing is selected. Fructose, glucose, and sucrose performed slightly better if a fast freezing speed was employed (Rules 31–36, Appendix A). Finally, an optimum cryoprotective performance could be obtained with medium concentrations (2.5%–12.5%) of trehalose and lactose. Furthermore, these disaccharides exhibited a similar behavior with both fast and slow freezing speeds (Rules 13–18 and 37–42, Appendix A).

Similar conclusions were obtained from the Δ PdI model set of rules. Although in general, a low Δ PdI was achieved, some differences among CPs were found. In the same way as reported for Δ Size, mannitol and sorbitol also exhibited a poor performance in terms of Δ PdI. However, the use of specific conditions, such as CP proportions ranging from 7.5% to 12.5% and a fast freezing speed, would allow us to obtain a low Δ PdI (Rules 31–50, Appendix A). Glucose, lactose, and fructose should be ideally employed in a low-medium concentration (up to 10%) to achieve a low Δ PdI. Regarding the freezing step, no relevant differences were found between freezing speeds for these three compounds (Rules 1–30, Appendix A). On the other hand, sucrose exhibited a different behavior, since it works better at medium-high concentrations and its concentration requirements vary depending on the freezing speed (above 3.75% and 12.5% for fast and slow freezing, respectively) (Rules 51–61, Appendix A). Lastly, the rules for trehalose indicated its use at low-mid proportions (up to 12.5% and in the range of 3.75%–12.5% for fast and slow freezing speeds, respectively) led to a small Δ PdI (Rules 62–71, Appendix A).

### 3.4. Influence of Cryoprotectant Properties and Operation Conditions over NLCs’ Characteristics (Model 2)

In a second and more detailed approach, the role of CP-specific characteristics (such as MW_CP_ and Π) and freezing speed on Δ Size, Δ PdI, and Δ ZP was modelled. FormRules^®^ (Intelligensys Ltd., Kirkwall, Scotland, UK) also succeeded in modelling Δ size and Δ PdI, leading to R^2^ higher than 70% and computed f values above the critical ones in both cases. A suitable model for Δ ZP was not found in this case. The information provided by the NFL software showed that both Δ Size and Δ PdI are explained by the interaction between MW_CP_ and Π. Moreover, Δ Size variations were also associated with the interaction of MW_CP_ and freezing speed, while changes in Δ PdI were attributed to the freezing speed (Table 3).

The rules for Δ Size model indicated CPs of low molecular weight, below 220 g/mol, performed better at low Π values. Moreover, those exhibiting a MW ranging from 220 to 300 g/mol needed medium–high Π values for obtaining low Δ Size. Interestingly, CPs with a MW above 300 g/mol showed a higher independence of Π values, as a low Δ Size would be obtained in all cases (Rules 1–9, Appendix A). Furthermore, similar results were obtained for Δ PdI, although, in this case, Π requirements were reported to increase progressively with the increase in MW_CP_ in all cases (Rules 1–9, Appendix A).

The interaction among MW_CP_ and freezing speed also played a role on Δ Size. In this way, CPs of MW below 220 g/mol performed better when fast freezing was employed. Slower freezing is advisable for CPs of MW ranging from 220 to 300 g/mol, while a higher independence of freezing speed was found for CPs showing a MW above 300 g/mol (Rules 10–15, Appendix A). As an example, fructose (MW_CP_ = 180.16 g/mol) at 10% (*w/v*) led to values of Δ Size of 47 ± 5 and 78 ± 7 nm when fast and slow freezing were carried out, respectively.

Furthermore, fast freezing speed promoted lower Δ PdI, as indicated by a lower membership degree for rule 11 (Rules 10–11 Appendix A).

## 4. Discussion

NLCs’ formulations demonstrated a good ability to endure the lyophilization process. This phenomenon might be related to the freezing speeds selected in this work (−80 °C and −196 °C), which are likely to limit nanoparticle movement, reducing aggregation [27]. However, LP exhibited a gummy-like nature and a challenging re-dispersion, probably due to the presence of a high residual water content along with the lack of a porous structure. These features are related to the collapse of the formulation structures [10] and justify the use of cryoprotectants.

CPs are usually employed during lyophilization to protect nanoparticles from freezing stress [28], reduce aggregation, and improve re-dispersion [27]. Several mechanisms of action of CPs have been proposed. Some authors have suggested that CPs generate a glassy matrix when the glass transition temperature of the maximum cryo-concentrated solution (Tg′) is reached, in which the nanoparticles are immobilized and protected [8,10,29]. Tg′ corresponds to the glass transition temperature (Tg) of the highly concentrated solution generated after the formation of ice crystals during freezing [10,29]. Other authors have suggested that, during freezing, nanoparticles could be isolated by sugar molecules in the unfrozen fraction, increased in volume by the addition of CPs, without requiring sugar vitrification [30]. In this second theory, the sugar acts as a scaffolding, inhibiting nanoparticle movement locally, as previously described for protein stabilization [31]. CPs can also act as lyoprotectants, conferring protection from drying stress [32], by generating hydrogen bonds with the polar groups of the nanoparticle surface, thus replacing water molecules [7,8,10].

In this work, in order to deepen the usefulness of carbohydrates as cryoprotectants for the lyophilization of NLCs, several of them widely used as CPs were selected, among which were sugar alcohols, monosaccharides, and disaccharides, and they were tested at different concentrations and conditions. After nanoparticle lyophilization and reconstitution, particle size, PdI, and ZP were determined and compared with their initial values. Highly heterogeneous values for both Δ Size and Δ PdI were obtained (Figure 2 and Figure 3), with several patterns for the different lyophilization conditions tested (freezing speed, CP type, and concentration). On the contrary, the zeta potential variations were small, although positive Δ ZP was observed for some CPs such as lactose, fructose, or trehalose (Figure 4). It is likely that this phenomenon is more related to the adsorption of CP molecules on the surface of the nanoparticles, due to interactions between these and the OH groups of the CPs, than to a real modification of the surface of the nanoparticle during the process of lyophilization [10,33].

Based on the results obtained from the NFL modelling (model 1), we proposed a traffic light system to establish the effect of CP, its concentration (%CP), and freezing rate on the reconstituted NLCs’ characteristics and help in the selection of CP and process conditions (Figure 5). Not desirable conditions to obtain low Δ size or Δ PdI are indicated in red while highly advisable conditions are denoted in green. This classification system was set up from the obtained “IF-THEN” rules and their membership degrees (MD). Green and red colors represent lyophilization conditions which, according to “IF-THEN” rules, exhibited MD values higher than 0.75 to obtain low or high Δ size/Δ PdI values. Yellow colors represent conditions leading to either low or high Δ size/Δ PdI values, obtained from rules showing a MD ranging from 0.5–0.74.

Interestingly, the optimal ranges of %CP proposed by the NFL software for sugars such as sucrose, fructose, glucose or trehalose were in agreement with those selected by other authors as having the best protective effect [34] (Figure 5).

The second NFL model, focused on understanding the role of certain CP parameters (such as MW_CP_ and Π, along with freezing speed) on LP properties, was in agreement with model 1, with a slightly lower predictive capacity.

Results from model 2 indicated the MW_CP_ determined the optimal freezing speed and Π to obtain a product with easy redispersion and adequate characteristics (low Δ Size and Δ PdI). Thus, for CPs of MW below 220 g/mol (monosaccharides and sugar alcohols), low Π values and a fast freezing speed are preferred. CPs with a MW above 300 g/mol (disaccharides) work better at high Π values, and, in these cases, the process was barely influenced by the freezing speed.

The scientific literature shows some controversy regarding the most suitable freezing speed for nanocarriers’ lyophilization [15,35]. Model 1 showed a general preference for fast freezing using liquid nitrogen, in agreement with previous theories, which indicated that fast supercooling leads to the generation of small ice crystals, reducing mechanical stress over nanocarriers [7,10]. Moreover, results from model 2 helped to explain this controversy, indicating that freezing speed requirements vary depending on the MW_CP_, in agreement with data reported by other authors [15]. As the freezing front progressed, ice crystals and a cryo-concentrated solution, consisting of nanoparticles and other formulation elements, were formed [10,15]. The number of CPs available to protect the nanoparticles varied as a function of the freezing speed and the diffusion rate of the CP molecules toward the cryo-concentrated phase [15], which, in turn, depended on the MW_CP_. Low MW CPs must migrate faster than high-molecular-weight ones.

In this way, the preference for a rapid freezing observed for CPs with a MW below 220 g/mol can be explained by their expected fast migration capacity. This feature could allow efficiently protecting NLCs and also to take advantage from the fast freezing benefits, as smaller ice crystals’ generation [10,15]. Meanwhile, NLCs’ dispersions stabilized with CPs exhibiting a MW ranging from 220 to 300 g/mol, exhibiting better properties in terms of Δ Size if a slower freezing rate was employed.

On the other hand, the higher independence from freezing speed observed for the biggest molecules can be related to their superior Tg [36]. Disaccharides have higher Tg than monosaccharides [37], which explains why they vitrify earlier during the freezing process, immobilizing the NLCs in a vitreous matrix and, thus, minimizing particle damage and the effects of freezing. This effect is likely to be responsible for the lower prediction capacity of the second NFL model, due to the involvement of other CP characteristics different than MW.

Furthermore, the variations in Π requirements as a function of the MW_CP_ described in the second model could also be explained by the diffusion phenomenon. Hence, the addition of a high number of poorly diffusive CPs to the nanoparticles could increase the presence of these compounds in the cryo-concentrated liquid phase and counteract their slow diffusion [15]. Additionally, considering Π is a colligative property, its requirements would be directly related with %CP, which explains the high similarity between the two NFL models.

As an example, fructose and glucose (MW_CP_ = 180.16 g/mol) were found to be more effective when employed up to a certain concentration, while the use of higher CP proportions is more advisable for sucrose (MW_CP_ = 342.3 g/mol). Nevertheless, trehalose and lactose (MW_CP_ = 342.3 g/mol) seemed to exhibit different behaviors than those described for sucrose, as they were found to perform better when employed at medium concentrations (Figure 5). These contradictory findings could be associated, on the one hand, to the remarkable protective activity of trehalose, triggered by features such as low hygroscopicity, higher glass transition temperature, or absence of internal hydrogen bonding [10,12,13,14]. On the other hand, lactose crystallization during freezing [38] could lead to the low %CP requirements observed. In this way, the formation of a eutectic with ice and the generation of CP crystals could favor nanoparticle aggregation and fusion [10] since the generation of these ice crystals might induce shear stresses on the nanoparticles and cause a loss of cryoprotectant–nanoparticle interactions [31]. Furthermore, mannitol and sorbitol have also been reported to crystallize during the lyophilization process [39], which could explain the poor cryoprotective effectivity achieved with both CPs.

Therefore, in the same way as described for freezing speed, crystalline behavior could also be associated with the lower prediction capacity shown by the second model, in comparison with the first one.

## 5. Conclusions

The neurofuzzy logic analysis allowed a better understanding of the role of lyophilization conditions, such as freezing speed or certain characteristics of the carbohydrate cryoprotectants, on the properties of the obtained nanoparticles and to provide some insights into the physicochemical phenomena involved. The knowledge generated allows a rational selection (avoiding trial and error approaches) of the variables used for lyophilization to obtain easily redispersible NLCs, with similar characteristics to those initially produced. NLCs’ lyophilization could be performed using a considerable variety of variables (CP choice, proportion employed, or freezing speed) as long as they are properly combined. In this way, the use of monosaccharides such as glucose or fructose in a concentration up to 10% and a fast freezing speed is highly advisable. Moreover, the addition of disaccharides, such as sucrose and trehalose, at concentrations higher than 12.5% and in the range of 3.75%–12.5%, respectively, can also constitute interesting options to obtain NLCs with suitable properties with any of the freezing speeds evaluated. Nonetheless, the usage of sugar alcohols, such as mannitol or sorbitol, would not be advisable.

## Figures and Tables

**Figure 1 pharmaceutics-13-01330-f001:**
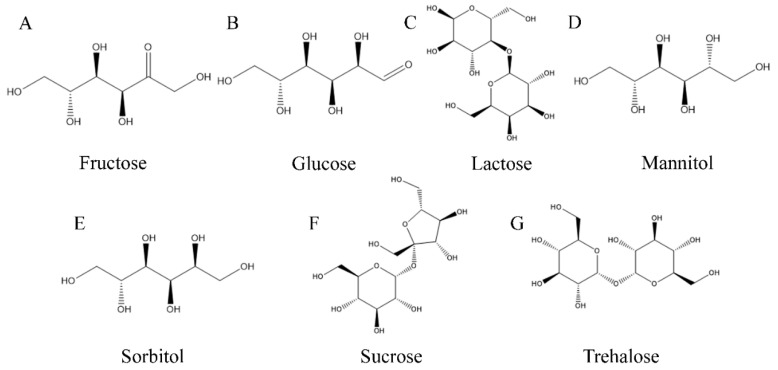
Chemical structure of (**A**) Fructose, (**B**) Glucose, (**C**) Lactose, (**D**) Mannitol, (**E**) Sorbitol, (**F**) Sucrose, and (**G**) Trehalose.

**Figure 2 pharmaceutics-13-01330-f002:**
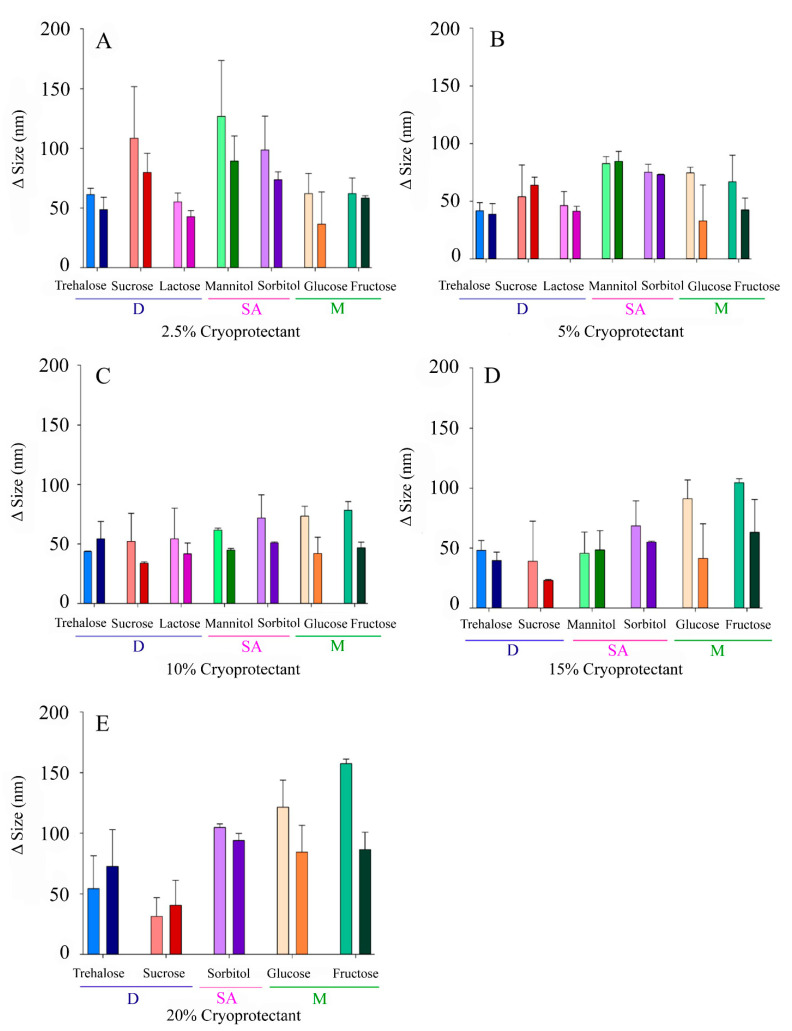
Increase in the sizes of the NLCs (Δ Size) after lyophilization with different CPs at variable concentrations: (**A**) 2.5% *w/v*, (**B**) 5% *w/v*, (**C**) 10% *w/v*, (**D**), 15% (*w/v*), (**E**) 20% *w/v*. Light colours correspond to slow freezing processes and dark colours to fast freezing processes. The CPs are grouped according to their type into disaccharides (D), sugar alcohols (SA), and monosaccharides (M).

**Figure 3 pharmaceutics-13-01330-f003:**
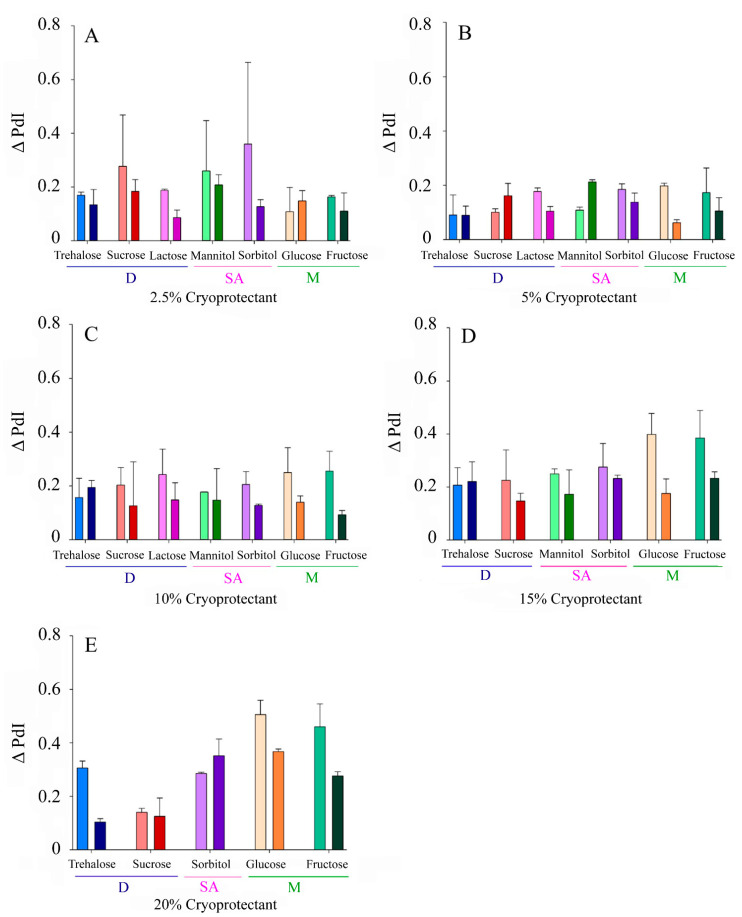
Increase in the polydispersity index of the NLCs (Δ PdI) after lyophilization with different CPs at variable concentrations: (**A**) 2.5% *w/v*, (**B**) 5% *w/v*, (**C**) 10% *w/v*, (**D**), 15% (*w/v*), (**E**) 20% *w/v*. Light colours correspond to slow freezing processes and dark colours to fast freezing processes. The CPs are grouped according to their type into disaccharides (D), sugar alcohols (SA), and monosaccharides (M).

**Figure 4 pharmaceutics-13-01330-f004:**
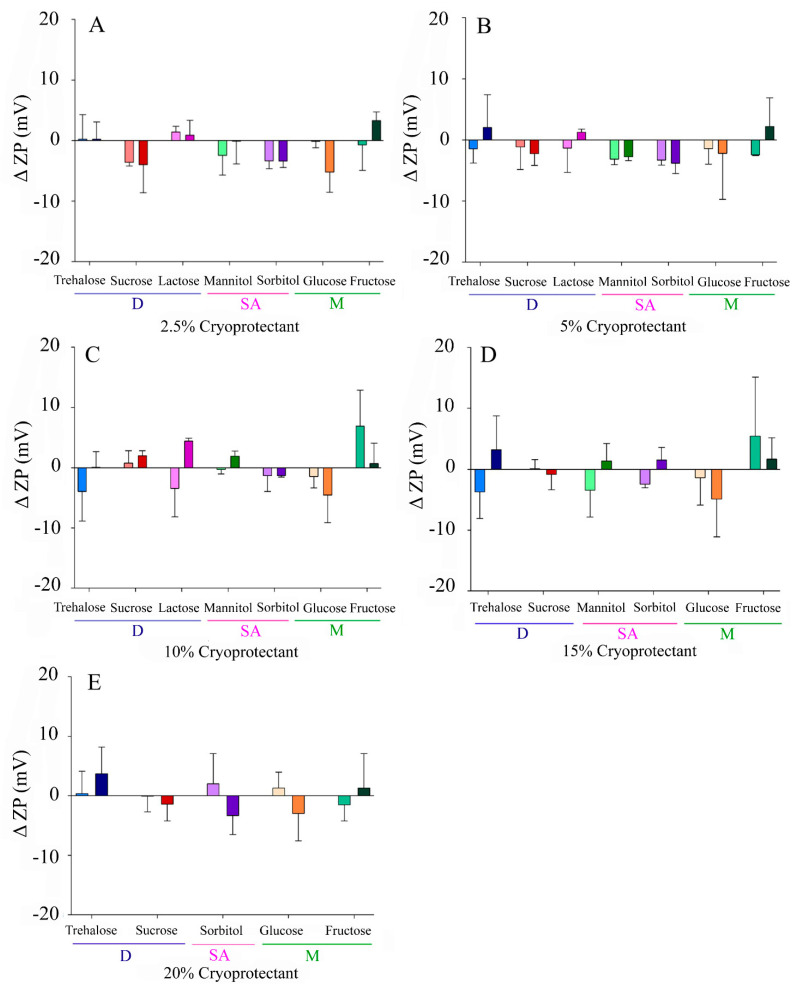
Increase in zeta potential of the NLCs (Δ ZP) after lyophilization with different CPs at variable concentrations: (**A**) 2.5% *w/v*, (**B**) 5% *w/v*, (**C**) 10% *w/v*, (**D**), 15% (*w/v*), (**E**) 20% *w/v*. Light colours correspond to slow freezing processes and dark colours to fast freezing processes. The CPs are grouped according to their type into disaccharides (D), sugar alcohols (SA), and monosaccharides (M).

**Figure 5 pharmaceutics-13-01330-f005:**
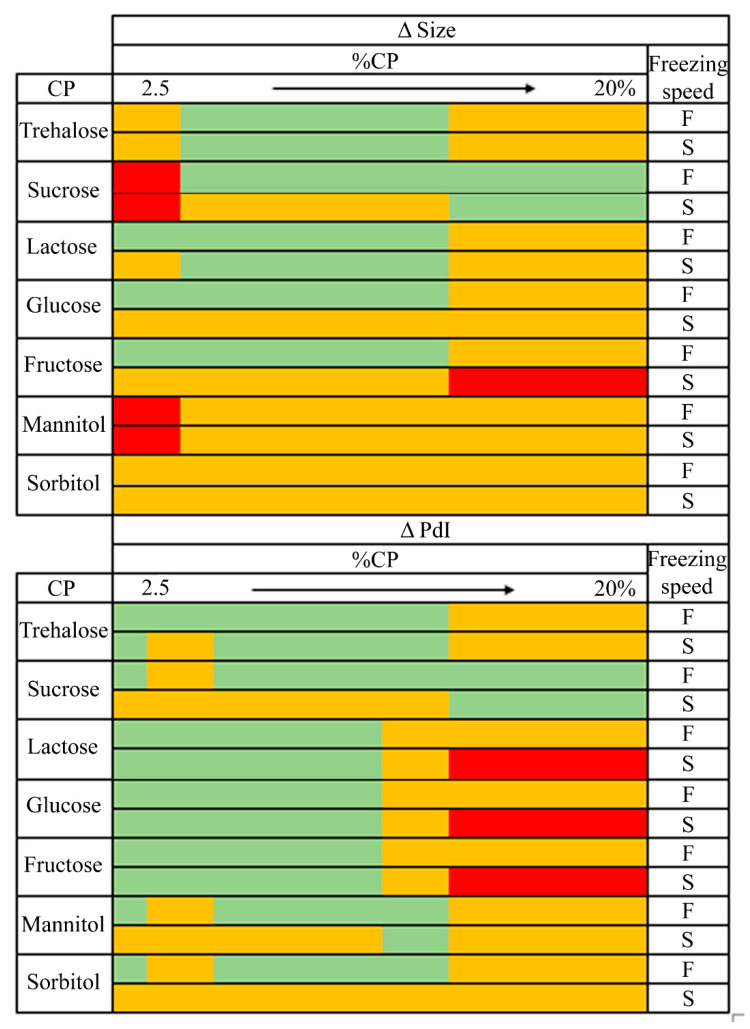
Traffic light system from NFL models for particle size and polydispersity index of NLCs. F: fast freezing speed. S: slow freezing speed. Highly advisable conditions in green, mediumly advisable in yellow, and not advisable in red.

**Table 1 pharmaceutics-13-01330-t001:** Variables included as inputs for modelling: CP type, CP concentration, MW_CP,_ and osmolarity in aqueous solution.

CP	CP Concentration (%)	Mw_CP_ (g/mol)	Π (mmol/kg)
Fructose	2.5	180.16	154
Fructose	5	180.16	279
Fructose	10	180.16	551
Fructose	15	180.16	894
Fructose	20	180.16	1141
Glucose	2.5	180.16	135
Glucose	5	180.16	257
Glucose	10	180.16	547
Glucose	15	180.16	820
Glucose	20	180.16	1127
Mannitol	2.5	182.17	142
Mannitol	5	182.17	270
Mannitol	10	182.17	593
Mannitol	15	182.17	945
Sorbitol	2.5	182.17	131
Sorbitol	5	182.17	253
Sorbitol	10	182.17	615
Sorbitol	15	182.17	864
Sorbitol	20	182.17	1146
Sucrose	2.5	342.3	73
Sucrose	5	342.3	140
Sucrose	10	342.3	300
Sucrose	15	342.3	434
Sucrose	20	342.3	602
Trehalose	2.5	342.3	70
Trehalose	5	342.3	134
Trehalose	10	342.3	281
Trehalose	15	342.3	381
Trehalose	20	342.3	538
Lactose	2.5	342.3	72
Lactose	5	342.3	142
Lactose	10	342.3	248

**Table 2 pharmaceutics-13-01330-t002:** Inputs selected by NFL models that explain Δ Size, Δ PdI, and Δ ZP in lyophilized NLCs’ formulations, along with predictability (R^2^) and ANOVA parameter (f: critical value of the f distribution). The most relevant submodel is bolded, while models not meeting quality criteria are highlighted in red.

Output	Submodels	Inputs from FormRules^®^	R^2^	Calculated f Value	Degrees of Freedom	f Critical for *p* ˂ 0.01
**Δ size**	Submodel 1	CP × Speed	91.77	10.17	34 and 31	2.32
	**Submodel 2**	**CP × %CP**				
**Δ PdI**	**Submodel 1**	**CP × %CP**	76.04	8.29	18 and 47	2.34
	Submodel 2	Speed				
	Submodel 3	%CP				
**Δ ZP**	**Submodel 1**	**CP × Speed**	51.35	3.52	15 and 50	2.42
	Submodel 2	%CP				

**Table 3 pharmaceutics-13-01330-t003:** Inputs selected by NFL models that explain Δ Size, Δ PdI, and Δ ZP in lyophilized NLCs’ formulations, along with predictability (R^2^) and ANOVA parameter (f: critical value of the f distribution). The most relevant submodel is bolded, while models not meeting quality criteria are highlighted in red.

Output	Submodels	Inputs from FormRules^®^	R^2^	Calculated f Value	Degrees of Freedom	f Critical for *p* ˂ 0.01
**Δ size**	**Submodel 1**	**MW_CP_ × Π**	74.38	10.14	14 and 49	2.47
	Submodel 2	MW_CP_ × Speed				
**Δ PdI**	**Submodel 1**	**MW_CP_ × Π**	70.50	12.65	10 and 53	2.68
	Submodel 2	Speed				
**Δ ZP**	**Submodel 1**	**Π**	1.58	0.49	2 and 61	4.97

## Data Availability

Dynamic light scattering measurements performed in this study are available in Appendix A.

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
