# Peer review of "A Traffic Light System to Maximize Carbohydrate Cryoprotectants’ Effectivity in Nanostructured Lipid Carriers’ Lyophilization"

_pharmaceutics, 2021, doi:10.3390/pharmaceutics13091330_

Round 1

Reviewer 1 Report

The authors in the present study investigated the application of a Neurofuzzy Logic (NFL) to understand the effect of cryoprotectants on the lyophilization process of nanostructured lipid carriers. The paper is interesting and I recommend the publication of the manuscript in the journal after addressing the following issues.

  1. Authors are advised to provide more details of the application of AI tools in the optimization of pharmaceutical processes. Authors have just listed the applications. I suggest to provide a summary of these publications (1-2 sentences), this can help the reader understand better.
  2. Why was the specific ratio of the components (solid lipid to liquiid lipid ratio) used for NLC's. Did the authors perform any optimization studies? or this ratio provided NLC's with minimum sizes? The reason for this question is that loading of an active ingredient can affect the size and ZP of the NLCs
  3. Why statistical analysis was not performed to understand the effect of the freezing process on the increase of size, PDI and ZP. Graphs show that there was no significant difference between slow and fast freezing on size and PdI (preliminary thought). It is advised to provide a conclusion based on a statistical analysis.
  4. It should have been interesting to see the effect of all the parameters using other techniques such as DSC or imaging (polarized microscopy). Only size, PdI and ZP is very basic and provides limited information.
  5. Evaluation of drug loaded NLCs should have added more weight to the paper (advise for extension of this paper).

Besides all these issues, the manuscript is well written and interesting. However, the investigation in current form is basic and requires some more experiments to better apply the AI tools in lyophilization process of NLCs. 

Author Response

We would like to thank the reviewer for the careful reading of the manuscript as well as for the positive feedback given. We have answered each comment below (our responses in blue) and we think the manuscript has notably been improved. Changes on the main manuscript and supplementary file have been highlighted in yellow.

Reviewer #1 comments:

Comment 1: Authors are advised to provide more details of the application of AI tools in the optimization of pharmaceutical processes. Authors have just listed the applications. I suggest to provide a summary of these publications (1-2 sentences), this can help the reader understand better.

We agree with the reviewer on the need of providing more information regarding the application of AI tools in pharmaceutical processes optimization. The following text has been added to the introduction section:

“In these systems, AI allowed to predict the endpoint of the granulation process, to obtain particulate drug delivery systems exhibiting optimal characteristics, and also to develop smart thermosensitive hydrogels [17-21]”

Comment 2: Why was the specific ratio of the components (solid lipid to liquid lipid ratio) used for NLC's. Did the authors perform any optimization studies? or this ratio provided NLC's with minimum sizes? The reason for this question is that loading of an active ingredient can affect the size and ZP of the NLCs.

The solid to liquid lipid ratio, along with other formulation parameters was optimized in a previously published research work from our lab: “Rouco, H.; Diaz-Rodriguez, P.; Rama-Molinos, S.; Remunan-Lopez, C.; Landin, M. Delimiting the knowledge space and the design space of nanostructured lipid carriers through Artificial Intelligence tools. Int J Pharm 2018, 553, 522-530, doi:10.1016/j.ijpharm.2018.10.058”. In this work, we selected the most suitable composition and operation conditions in order to obtain nanoparticles with optimal properties, by applying Artificial Intelligence tools, in the same way as in the present paper.

To clarify this point the following sentence has been added to the materials and methods section: NLC formulations were prepared by hot high shear homogenization, following the procedure and composition previously optimized in our laboratory through Artificial Intelligence tools [19].

Comment 3: Why statistical analysis was not performed to understand the effect of the freezing process on the increase of size, PDI and ZP. Graphs show that there was no significant difference between slow and fast freezing on size and PdI (preliminary thought). It is advised to provide a conclusion based on a statistical analysis.

In this work, we propose the use Artificial Intelligence tools such as Neurofuzzy Logic as an alternative to traditional approaches based on statistical analysis. We agree with the reviewer on the existence of high standard deviations in some of the experimental data shown in the graphs. However, in order to avoid misinterpretations related with these experimental variations, we have applied Structural risk minimization as model selection criteria (as specified in the materials and methods section), to minimize the interference of experimental deviations over the conclusions gathered from the Neurofuzzy Logic models. Finally, we applied statistical analysis (ANOVA) to evaluate the differences between experimental and predicted data, in order to check the validity of the developed models. In this way, the software allowed us to detect trends between inputs and outputs despite the existence of experimental errors. This approach avoids the need of performing any additional statistical analysis.

Comment 4: It should have been interesting to see the effect of all the parameters using other techniques such as DSC or imaging (polarized microscopy). Only size, PdI and ZP is very basic and provides limited information.

We agree with the reviewer that the use of DSC and imaging techniques could provide further information regarding the effect of the analyzed parameters over lyophilized NLC properties. However, the aim of this work is to point out the critical parameters during lyophilization analyzing several cryoprotectants at variable concentrations, therefore, a large number of formulations was obtained making difficult to perform these techniques for all of them.

On the other hand, in future works, a further characterization of the lyophilized powders using the most promising lyophilization conditions will be performed.

Comment 5: Evaluation of drug loaded NLCs should have added more weight to the paper (advise for extension of this paper).

We agree with the reviewer on the fact that the performance of the present lyophilization study using drug loaded NLCs could have been given more strength to the paper. However, the main reasons behind the selection of blank nanoparticles were both economical and environmental. In our laboratory, we are working with an anti-mycobacterial compound which is expensive and also makes necessary a careful disposal of the generated wastes. In this way, and in line with the reviewer´s opinion, our idea is to apply in future works only the most promising lyophilization conditions (according to the knowledge generated in the present work) to lyophilize our drug-loaded NLC formulations.

Reviewer 2 Report

The article is well written, and the study is well planned, with interesting results for researchers involved in the development of Nanostructured Lipid nanoparticles. However, please provide DLS measurements data to supplementary materials as the quality of these measurements is of key importance for this study, and readers should be allowed to see these measurements. 

Author Response

We would like to thank the reviewer for the careful reading of the manuscript as well as for the positive feedback given. We have answered each comment below (our responses in blue) and we think the manuscript has notably been improved. Changes on the main manuscript and supplementary file have been highlighted in yellow.

Comment 1: The article is well written, and the study is well planned, with interesting results for researchers involved in the development of Nanostructured Lipid nanoparticles. However, please provide DLS measurements data to supplementary materials as the quality of these measurements is of key importance for this study, and readers should be allowed to see these measurements.

DLS measurement data have been added to the supplementary materials file, as suggested by the reviewer.

Reviewer 3 Report

The submitted paper deals with the effects of Lyophilization conditions on the technological features of NLCs. A wide experimental campaign has been carried out aiming to find out how to select a proper nature and amount of the CP. The outcomes indicate that CP molecular weight is a major parameter indicating freezing conditions and CP amount.  The criterion based on NFL is pretty interesting, and holds some potential for possible industrial applications and for the determination of critical quality attributes and in a quality-by-design approach.

The paper is well written and organized. Check for some typos (e.g. “stablish” in line 66 should be “established”; please, verify throughout the article).

The main issue with the paper is an insufficient description of AI-NFL approach which should be more clearly presented for the reader’s convenience.

The bibliography is pretty outdated (only 8 references are from 2016 on). More recent papers should be cited.

Author Response

We would like to thank the reviewer for the careful reading of the manuscript as well as for the positive feedback given. We have answered each comment below (our responses in blue) and we think the manuscript has notably been improved. Changes on the main manuscript and supplementary file have been highlighted in yellow.

Comment 1: The paper is well written and organized. Check for some typos (e.g. “stablish” in line 66 should be “established”; please, verify throughout the article).

Typographical errors have been revised and corrected as requested by the reviewer.

Comment 2: The main issue with the paper is an insufficient description of AI-NFL approach which should be more clearly presented for the reader’s convenience.

We agree with the reviewer on the need of carefully describing the NFL approach. In this way, a more detailed description of the NFL software operation has been included in the introduction and on the materials and methods section. The added texts can be found below:

“The main unit of an artificial neural system is the artificial neuron or node, these nodes establish connections with each other and the strength of these connections is known as weight [22]. However, the interpretation of these ANN might not be a simple task [23].”

“Neurofuzzy logic systems need to be trained to learn from data, and during this training process they establish relationships between the inputs and the outputs of the database using different algorithms to alter the strength of the connections in the neural network [22]. This process allows to modulate the signal flow, establishing generalizations be-tween the variables studied (inputs) and the nanoparticle characteristics (outputs) [22].”

References included:

“Landin, M.; Rowe, R.C. Artificial neural networks technology to model, understand, and optimize drug formulations. In: Formulation Tools for Pharmaceutical Development; Elsevier: 2013; pp. 7-37.”

“Colbourn, E.A.; Rowe, R.C. Neural Computing Boosts Formulation Productivity. Pharmaceutical Technology 2003, 27, 22-25.”

Comment 3: The bibliography is pretty outdated (only 8 references are from 2016 on). More recent papers should be cited.

We have updated the references adding recent works relevant of the field:

“Franze, S.; Selmin, F.; Samaritani, E.; Minghetti, P.; Cilurzo, F. Lyophilization of Liposomal Formulations: Still Necessary, Still Challenging. Pharmaceutics 2018, 10, doi:10.3390/pharmaceutics10030139”.

“Khan, A.A.; Abdulbaqi, I.M.; Abou Assi, R.; Murugaiyah, V.; Darwis, Y. Lyophilized Hybrid Nanostructured Lipid Carriers to Enhance the Cellular Uptake of Verapamil: Statistical Optimization and In Vitro Evaluation. Nanoscale Res Lett 2018, 13, 323, doi:10.1186/s11671-018-2744-6.”

“Amis, T.M.; Renukuntla, J.; Bolla, P.K.; Clark, B.A. Selection of Cryoprotectant in Lyophilization of Progesterone-Loaded Stearic Acid Solid Lipid Nanoparticles. Pharmaceutics 2020, 12, doi:10.3390/pharmaceutics12090892.”

“Patel, P.; Patel, M. Enhanced oral bioavailability of nintedanib esylate with nanostructured lipid carriers by lymphatic targeting: In vitro, cell line and in vivo evaluation. Eur J Pharm Sci 2021, 159, 105715, doi:10.1016/j.ejps.2021.105715.”

“Khan, A.A.; Mudassir, J.; Akhtar, S.; Murugaiyah, V.; Darwis, Y. Freeze-Dried Lopinavir-Loaded Nanostructured Lipid Carriers for Enhanced Cellular Uptake and Bioavailability: Statistical Optimization, in Vitro and in Vivo Evaluations. Pharmaceutics 2019, 11, doi:10.3390/pharmaceutics11020097.”

“Trenkenschuh, E.; Friess, W. Freeze-drying of nanoparticles: How to overcome colloidal instability by formulation and process optimization. Eur J Pharm Biopharm 2021, 165, 345-360, doi:10.1016/j.ejpb.2021.05.024.”

Round 2

Reviewer 1 Report

Authors have addressed all my comments and updated the manuscript accordingly. The manuscript can be accepted for publication.